# Children with Cerebral Palsy Have Similar Walking and Running Quality Assessed by an Overall Kinematic Index

**DOI:** 10.3390/ijerph18094683

**Published:** 2021-04-28

**Authors:** Devin K. Kelly, Mark L. McMulkin, Corinna Franklin, Kevin M. Cooney

**Affiliations:** 1Kinesiology and Nutrition Sciences, University of Nevada, Las Vegas, NV 89154, USA; devin.kelly@unlv.edu; 2Shriners Hospitals for Children®, Spokane, WA 99204, USA; 3Shriners Hospitals for Children®, Philadelphia, PA 19140, USA; cfranklin@shrinenet.org; 4Shriners Hospitals for Children®, Erie, PA 16505, USA; kcooney@shrinenet.org

**Keywords:** cerebral palsy, running, gait deviation index, running kinematics, gait

## Abstract

Running ability is critical to maintaining activity participation with peers. Children and adolescents with cerebral palsy (CP) are often stated to run better than they walk, but running is not often quantitatively measured. The purpose of this study was to utilize overall gait deviation indices to determine if children with diplegic CP run closer to typically developing children than they walk. This retrospective comparative study utilized 3D running kinematics that were collected after walking data at two clinical motion analysis centers for children with diplegic cerebral palsy. Separate walking and running Gait Deviation Indices (GDI Walk and GDI* Run), overall indices of multiple plane/joint motions, were calculated and scaled for each participant so that a typically developing mean was 100 with standard deviation of 10. An analysis of variance was used to compare the variables Activity (walking vs running) and Center (data collected at two different motion analysis laboratories). Fifty participants were included in the study. The main effect of Activity was not significant, mean GDI Walk = 76.4 while mean GDI* Run = 77.1, *p* = 0.84. Mean GDI scores for walking and running were equivalent, suggesting children with diplegic cerebral palsy as a group have similar walking and running quality. However, individual differences varied between activities, emphasizing the need for individual assessment considering specific goals related to running.

## 1. Introduction

Cerebral palsy (CP) refers to a group of permanent, non-progressive disorders which are a result of an insult to the immature brain. The motor function of children with CP varies considerably, with some children experiencing functional and activity limitations [1,2]. Regardless of functional limitations, running is an important activity of daily living for all children that offers physical and psychosocial health benefits. For children with CP to experience the benefits of running, clinicians might evaluate both walking and running for treatment planning. 

It has been observed by clinicians who treat individuals with CP that they appear to run better than they walk [3]. The publication “The Boy Who Could Run But Not Walk” describes running as a skill acquired later in development, while walking is developed by the damaged, immature brain [3]. Therefore running patterns in individuals with CP are closer to those observed in typically developing children compared to walking patterns. The concept of running patterns in children with CP appearing closer to typically developing children in this publication is largely based on qualitative reports of gait observation. 

Quantitative three-dimensional kinematic analysis of walking is well established as a useful tool for evaluation and treatment planning for children with CP, while the inclusion of kinematic analysis of running for this purpose is a less common practice. There is discord in conclusion about the running ability of children with CP in the literature. Improvements in running compared to walking including resolution of sagittal plane abnormalities have been reported [4,5]. Davids et al. examined kinematics and kinetics of walking and running in children with diplegic CP and determined that running was more typical than walking [4]. However, this study relied on point data such as peak values or a wave-form analysis that focused on single joints and only the sagittal plane instead of a measure of overall kinematics. Children with unilateral CP have been reported to improve in running, utilizing an overall classification system specifically for children with unilateral CP based on sagittal plane kinematics [5]. Contrasting studies have reported running as less typical for children with CP compared to walking [6]. Böhm and Döderlein examined gait symmetry (magnitude of absolute difference between limbs) in spatiotemporal parameters, kinematics, and kinetics of walking and running in children with unilateral and bilateral CP [6]. They report overall lower symmetry for running including peak hip extension, knee flexion during loading, and ankle dorsiflexion. 

Several characteristics have been reported to be altered in running for children with CP. During running, children with CP have been shown to have reduced leg stiffness [7], altered dynamic stability (increased medio-lateral margin of stability) [8], and reduced ankle plantarflexor power compared to typically developing children [9]. Further, a running training program has been shown to increase ankle power generation and hip flexor power in swing [10]. These publications referencing altered characteristics in running for children with CP are important, but do not compare the overall kinematic quality between walking and running.

Previous literature provides mixed evidence regarding whether children with CP show fewer deviations compared to typically developing children for running compared to walking, relying on point data such as peak values, single joints, and the sagittal plane, rather than an overall measure of kinematics of the entire gait cycle. It may not be suitable to generalize conclusions about similarity in the quality of complex movement based on singular variables. A solution is the inclusion of the Gait Deviation Index (GDI) [11]. The GDI is used to describe overall gross motor function during walking that considers multiple variables in all three planes. A similar global index of running quality can be implemented in gait analyses if a center has available kinematic running data for typically developing children. Clinical evaluation of walking and running using gait deviation indices could provide insight into gross motor function in those with movement disorders such as CP.

The purpose of this study is to utilize overall gait deviation indices to determine if children with diplegic CP run closer to typically developing children than they walk. The null hypothesis tested is that an overall gait deviation index of running will be equivalent to the gait deviation index of walking. If this hypothesis is accepted, it implies that the magnitude of difference in gait quality between children with CP and typically developing children is similar between walking and running. This article is organized by presenting the method to collect kinematic walking and running data for children with CP, followed by description of GDI calculations of overall walking and running movement. Analysis and results focus on the overall measure of multiple joints and planes of motion, the GDI.

## 2. Materials and Methods

### 2.1. Participants

Approval was obtained from the Western Institutional Review Board for this retrospective multi-site study conducted at two Shriners Hospitals for Children centers (Erie, PA and Spokane, WA, USA). Running kinematics for children with CP have been collected alongside walking data used for clinical gait analyses at both centers. A sample of convenience was used to identify up to the most recent 30 children at each center meeting the inclusion criteria: (1) ages 4–21 at completion of gait analysis, (2) diagnosis of spastic diplegic cerebral palsy, (3) categorized as Gross Motor Function Classification System level (GMFCS) I or II [12], (4) no orthopedic surgery 2 years prior to gait analysis study, (5) no botulinum toxin injections 6 months prior to gait analysis, (6) no history of rhizotomy or baclofen pump, (7) ability to run barefoot with a verified flight phase from kinetic data. Verified flight phase was categorized as both limbs lacking contact with the ground for at least 0.01 s. 

### 2.2. Data Collection 

Three-dimensional kinematic and kinetic data were collected as participants walked and ran barefoot the length of the laboratory (40 feet), at their preferred speeds. In Erie, kinematic and kinetic data were collected at 120 Hz and 1200 Hz respectively for walking and running utilizing a 10-camera Motion Analysis Corporation (Santa Rosa, CA, USA) system and three AMTI (Watertown, MA, USA) force platforms. Data were processed using Visual 3D software (Germantown, MD, USA) to determine lower extremity joint angles for each task. Joint angles were obtained using a six degree of freedom model including 4 markers on pelvis, clusters of 4 markers on the thigh and shank segments, foot, and ankle markers [13]. Trajectories were filtered with a 4th order zero lag Butterworth, cutoff 10 Hz. Joint angle data were normalized to 0–100% of the gait cycle. Three successful trials (foot landing completely within the bounds of the force platform) per limb were collected for walking and five per limb for running. Force plate data were used only for detection of foot strike and toe off events in this study.

In Spokane, kinematic and kinetic data were collected at 100 Hz and 1000 Hz respectively for walking and running using a 12-camera Vicon system (Oxford, UK) and four AMTI force platforms. Vicon Plug-in-Gait was used to determine lower extremity joint angles for each task. The Plug-in-Gait model was modified by calculating the pelvic angles using a rotation, obliquity, tilt sequence [14]; in addition, patella [15] and tibial crest markers were employed instead of wands. Trajectories were also filtered with a 4th order zero lag Butterworth, cutoff 10 Hz. Joint angle data were to 0–100% of the gait cycle. Six successful trials (foot landing completely within the bounds of the force platform) per limb were collected for walking and one per limb for running.

### 2.3. Gait Deviation Index for Walking and Running

The GDI for walking is a tool to assess overall gait pathology which is scaled for typically developing walking to have a mean of 100 and standard deviation of 10 [11]. This index is based on the entire gait cycle for 9 kinematic variables in all planes of motion (pelvic tilt, pelvic rotation, pelvic obliquity, hip flexion/extension, hip rotation, hip ab/ad-duction, knee flexion/extension, ankle dorsi/plantar-flexion, foot progression). The GDI score calculated for a participant is a measure of overall gait deviation compared to a lab’s kinematic dataset for typical developing children. For activities other than walking (such as running or stair climbing), a derived GDI* score can be calculated when averaged kinematic data are available for the activity from persons with no pathology [16]. 

Clinical gait analysis centers at Shriners Hospitals for Children, Erie and Spokane participated in this study. Both centers had previously collected walking and running kinematics for typically developing children (under Institutional Review Board approved studies) to be used as a comparative dataset for clinical evaluation, planning, and treatment. These formed the basis for establishing GDI scores for walking and running. For Erie, the comparative dataset consisted of different participant pools for walking (*n* = 25, age 6–16) and running (*n* = 27, age 4–18). For Spokane, the same sample was used for both walking and running (*n* = 27, age 5–21 years). The typically developing participants are not the subjects for this study, but data from this group were utilized to determine GDI walking and running measures. 

The Gait Deviation Index for walking (GDI Walk) was determined at each location using center specific typically developing control participants as described in the GDI development [11]. The Gait Deviation Index for running (GDI* Run) was derived by starting with the calculation of the Gait Profile Score (GPS) [16] using mean running kinematics collected at each center from typically developing participants. The running GPS was then transformed to GDI* Run based on the strong relationship between the measures. The GDI* Run has a mean 100 score and standard deviation 10. The transformation from GPS to GDI as previously described [17] is shown in Appendix A. 

### 2.4. Data and Statistical Analyses

For each participant with CP, data for one walking trial was randomly selected for each limb (from trials collected at Erie and Spokane) and was used to determine GDI Walk. For running, one randomly selected trial for each limb from five running trials collected at Erie, and one running trial for each limb at Spokane, were used to determine GDI* Run. GDI scores for each limb were averaged to determine single overall GDI Walk and GDI* Run scores for each participant. 

Data were analyzed with a two factor repeated measures ANOVA. The first factor was Activity with two levels (GDI Walk and GDI* Run) and was a repeated measures variable. The second factor was Center with two levels (Erie and Spokane) and was a between subjects variable. The calculation of GDI Walk and GDI* Run were determined on the same participant with CP. The interaction of Center by Activity was also analyzed. Statistical significance threshold was set at *p* < 0.05 for comparing means between Activity and Center. 

## 3. Results

Data for 50 participants were included in this study (age range 4 to 17 years). Participant characteristics are described in Table 1. Mean self-selected walking speed was 1.10 m/s and mean running speed was 2.77 m/s.

The main effect of Activity was not significant (*p* = 0.84) indicating no significant difference between mean GDI Walk (76.4) and mean GDI* Run (77.1) values. The main effect of Center was significant (*p* = 0.001) indicating overall higher GDI scores at the Erie Center compared to the Spokane Center. The interaction of Center by Activity was found to be significant (*p* = 0.03). However, Student-Newman-Keuls post-hoc tests of the Center by Activity interaction (Figure 1) indicated that the only significant interactions were between confounded comparisons (such as GDI Walk at Erie compared to GDI* Run at Spokane). Therefore, main effects could be interpreted.

Example kinematic graphs for two subjects from each center along with site-specific control walking and running kinematics are shown in Appendix B. These figures help illustrate deviations across joints and planes that contribute to overall GDI walk and GDI*Run scores. In assessing strictly running kinematics, there was little consistency from participant to participant (Figure 2 shows the left side only of the 30 subjects from the Spokane center). In the sagittal plane, for hip and knee flexion/extension, many subjects show increased hip or knee flexion, but many also lacked hip/knee flexion in loading (also some have lack of knee flexion in swing). One reasonably common deviation was increased plantarflexion at the ankle. In the transverse plane, hip rotation and foot progression angle have a wide range from internal to external. There were not consistent running kinematic patterns for subjects with diplegic CP. Subjects with CP varied from typical running kinematics in both directions above and below typical curves (Figure 2).

To determine the effect of age on walking and running quality, a linear correlation between GDI Walk and GDI* Run was determined (Appendix C). There was a very poor correlation (low R^2^, <5% of variance in GDI explained by age) between age and both GDI scores. This indicates that older children with CP did not run or walk better than younger children.

## 4. Discussion

The purpose of this study was to utilize overall gait deviation indices to determine if children with diplegic CP run closer to typically developing children than they walk considering multiple joints and planes of motion. The null hypothesis that GDI* Run will be equivalent to GDI Walk was accepted. Mean GDI scores between the conditions varied by less than 1 point, which was not significantly different. This implies that, in general, the overall quality of running for children with CP was not different from walking. Mean GDI scores were different between centers, indicating overall higher gait and running quality for the participants included from the Erie center. Yet, two centers using different motion capture systems and different models each using its own control walking and running dataset found the same result of no mean difference between walking and running kinematic quality for children with diplegic CP, adding strength to the findings.

The study included children who were GMFCS level I (ability to perform gross motor skills including running) and II (minimal ability to perform gross motor skills including running) [12,18]. In retrospectively searching for up to the last 30 children with diplegic CP that underwent clinical gait analysis, five out of 25 participants at Erie, (three GMFCS Level I and two Level II) and three out of 33 participants at Spokane (all GMFCS level II) did not meet the criteria for running and were excluded from the study. For children and adolescents rated at GMFCS level II mobility, it can be expected that not all participants can run since this category was defined as having minimal ability to run and jump [12]. The prevalence of running ability has been reported as 55% of children and adolescents with diplegic CP rated at GMFCS level II [18]. In practice, children at each center who did not achieve running upon visual assessment during their gait analysis did not have running data collected at all, regardless of GMFCS level. Considering these exclusions and previous work on runners and non-runners [18], there is a potential for more children with CP at GMFCS levels I and II that do not have running ability. This supports our finding that children with diplegic CP do not run closer to typically developing children than they walk because they may not achieve running at all when quantitatively measured. 

Davids et al. analyzed walking and running in children with diplegic CP compared to typically developing children, similar to the present study [4]. The sagittal plane waveform analysis revealed ankle kinematics in children with diplegic CP and typically developing controls to be more similar during running as compared to walking. However, hip and knee kinematic waveforms were found to be similar between the two groups for both walking and running. The authors suggested that the observation that children with diplegic CP ran closer to typical than they walked came from the ability of the ankle to better tolerate deviations at higher speeds coupled with less affected proximal musculature allowing the hip to drive motion. The method used in the present study considers the differences in kinematic profiles across the entire gait cycle, and along with concurrent motion in multiple planes and joints. The inclusion of multiple planes of analysis and calculation of GDI to give an overall impression of gait for both walking and running may contribute to the difference in conclusions. 

This study had several limitations. The age range of the control dataset and participants spans a somewhat large range from 4 to 21 years of age. This was based upon the clinical population treated at the centers. Although mature gait has been suggested to occur at about age 5 [19], the age of development of mature running gait remains unknown. Running was collected barefoot following standard barefoot walking data collection. Able body control running data at both centers were collected barefoot so the comparison was equivalent. The length of both motion analysis centers was 40 feet which may have reduced running speed due to limited distance. Again, the same distance was used for participants with CP and typically developing participants that established the comparison data. The average running speed for able body control data used as a basis in this study (3.5 m/s and 3.2 m/s for each center) was equivalent to previous running speeds reported for adults during running kinematic data collection (3.2 m/s) [20]. Running ability might be impacted by balance [8] and confidence; assessments of these factors were not collected during the clinical visit.

The clinical implication of the findings of this study should be considered with treatment planning. At times, children and families express goals related to maintaining or improving running ability. Quantitative data on running kinematics might further elucidate anticipated outcomes and guide treatment decisions for these children with CP. Although there are no group differences between overall running and walking kinematics, individual running patterns varied emphasizing individual assessment (Figure 2). None of the participants had orthopedic surgical intervention within 2 years of the collected data as an inclusion criterion so effects of surgery were minimized with respect to running quality. While changes in walking kinematics due to orthopedic surgery are reported, previous studies have not focused on change in similar data for running kinematics. Orthopedic surgery intended to address walking deviations may have a different effect on the child’s ability to run but currently are largely unknown and should be an area of future research. Assessment of running data, along with walking for clinical treatment planning may be important for children and families to consider prior to orthopedic intervention. 

## 5. Conclusions

It has been perceived that individuals with CP run closer to typical than they walk. However, limited data are available that quantitatively compare three-dimensional joint angles of walking and running for the population of children with CP. The results of the present study focused on an index of overall movement kinematics. The overall index of three-dimensional joint angles showed that children with diplegic CP do not run closer to typically developing children than they walk. The age of the subjects participating in this study (4 to 17 years) did not affect the overall index for running or walking. These findings suggest that children with diplegic CP have similar movement quality for walking and running. 

## Figures and Tables

**Figure 1 ijerph-18-04683-f001:**
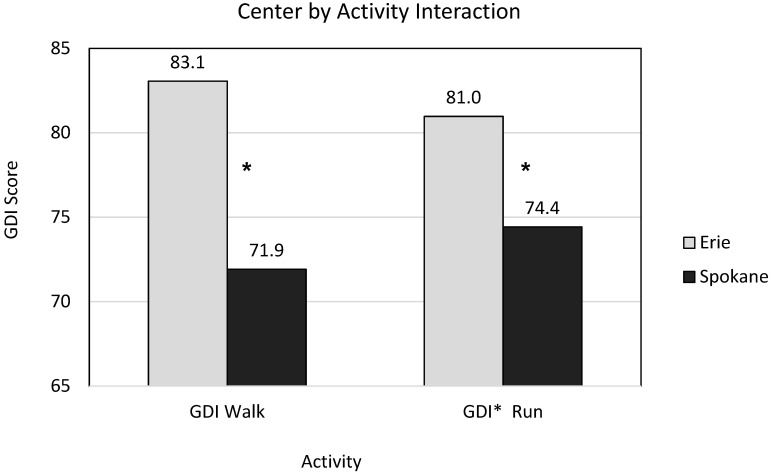
Comparison of Gait Deviation Index (GDI) for walking vs running for children with diplegic Cerebral Palsy at each Center. GDI Walk and GDI* Run were not significantly different from each other at either center. Both GDI Walk and GDI* Run were significantly higher at Erie compared to Spokane indicated by * using Student-Newman-Keuls post-hoc test.

**Figure 2 ijerph-18-04683-f002:**
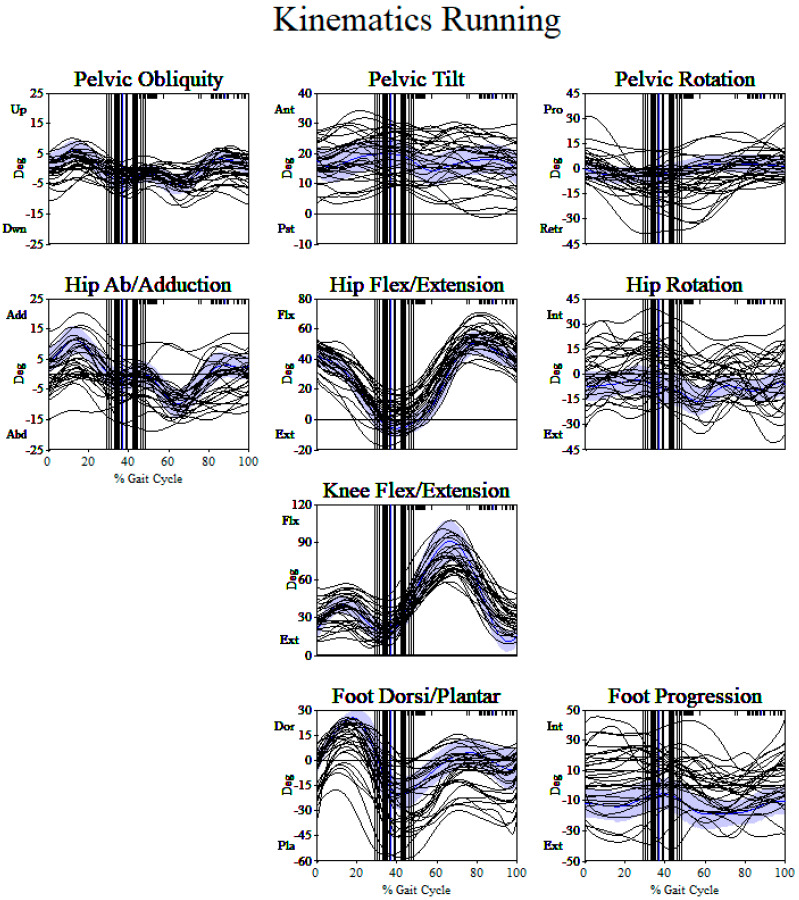
Running kinematics (left side only) for 30 participants from the Spokane center. Spokane center typically developing running kinematics are shown in shading. There are no consistent patterns of running from participant to participant. Variations from typically developing running are seen in both directions (e.g., increased flexion and extension in sagittal plane, increased internal and external rotation in the transverse plane).

**Table 1 ijerph-18-04683-t001:** Participant characteristics. Characteristics between Centers (Erie and Spokane) not significantly different, *p* > 0.05. Values are given as mean ± standard deviation.

Characteristic	Erie	Spokane	All
Sample Size (number)	20	30	50
GMFCS I (number)	11	17	28
GMFCS II (number)	9	13	22
Height (meters)	1.39 ± 0.18	1.45 ± 0.19	1.42 ± 0.18
Mass (kilograms)	36.9 ± 16.6	38.9 ± 15.1	38.1 ± 15.6
Age (years)	10.5 ± 3.2	11.6 ± 3.3	11.1 ± 3.3
range	4 to 16	5 to 17	4 to 17
Preferred Speed—Walk (meters/second)	1.16 ± 0.19	1.06 ± 0.15	1.10 ± 0.17
range	0.82 to 1.47	0.72 to 1.32	0.72 to 1.47
Preferred Speed—Run (meters/second)	2.71 ± 0.53	2.82 ± 0.67	2.77 ± 0.60
range	1.52 to 3.51	1.45 to 4.40	1.45 to 4.40

GMFCS—Gross Motor Functional Classification System.

## Data Availability

The data presented in this study are not available. The authors do not have permission to share data from their affiliated organizations.

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
