# Peer review of "Children with Cerebral Palsy Have Similar Walking and Running Quality Assessed by an Overall Kinematic Index"

_ijerph, 2021, doi:10.3390/ijerph18094683_

Round 1
Reviewer 1 Report
This interesting paper shows that CP patients have no significant difference in their development of walking and running when compared to their peers. The study is comprehensive and generally complete, however I believe there are some very important aspects of the paper missing:
1) I believe there should be an analysis based on age - surely older participants would have more experience in running?
2) I believe that the background sporting activities of the participants will also influence outcomes. In their conclusions the authors recommend "maintaining or improving running ability", but there is no clear evidence that this is beneficial - perhaps this is regarded as "common knowledge".
3) Both running and walking require confidence and balance. Some form of assessment of these attributes might support their final recommendations.
Minor comments:
1) The abstract mentions Activity and Centre. There is no clear outline of what is meant by "Centre" in the abstract.
2) p 4: The last sentence starting "There were not clear cut running ....". This is quite strange as the analysis is performed on the gait. If there is no clear pattern, how did the researchers decide on their segmentation for analysis? This must be explained.
Reviewer 2 Report
The authors present a manuscript comparing the deviation in walking verses running in children with cerebral palsy compared to typically developing children. The premise for comparison of walking verses running function is of interest to the gait analysis community specifically in treatment planning for children with cerebral palsy. The article is generally well written and easy to follow.
My main query to the authors is over their choice to analysis the data via the GDI rather than the GPS. Given that the GPS as the authors state can be calculated for any task if normative data is available and given that to calculate the GDI* Run the GPS was calculated and the GDI* Run was calculated based on the relationship between to the two summary scores. In addition when calculating the GPS the gait variable scores (GVS) is calculated for each of the 9 kinematic variables used in the GPS calculation allowing further analysis regarding deviation buy joint and plane which the authors have done so in a qualitative fashion. It appears the GVSs and GPS would be a more suitable summary score to use in this analysis.
I have a number of other comments contained in the annotated pdf

Author Response
Reviewer #2:
Comments and Suggestions for Authors
The authors present a manuscript comparing the deviation in walking verses running in children with cerebral palsy compared to typically developing children. The premise for comparison of walking verses running function is of interest to the gait analysis community specifically in treatment planning for children with cerebral palsy. The article is generally well written and easy to follow.
My main query to the authors is over their choice to analysis the data via the GDI rather than the GPS. Given that the GPS as the authors state can be calculated for any task if normative data is available and given that to calculate the GDI* Run the GPS was calculated and the GDI* Run was calculated based on the relationship between to the two summary scores. In addition when calculating the GPS the gait variable scores (GVS) is calculated for each of the 9 kinematic variables used in the GPS calculation allowing further analysis regarding deviation buy joint and plane which the authors have done so in a qualitative fashion. It appears the GVSs and GPS would be a more suitable summary score to use in this analysis.
Response: The comments are correct that the overall GDI* Run is derived from the GVS values for each of the 9 kinematic variables. It is possible to analyze each of the 9 GVS scores between running and walking. However, the main approach in this study was to provide a single overall measure of running quality. This could certainly be the GPS. However, the scaling of the GDI (mean typical 100 with standard deviation of 10) is felt to be a more easily understood variable. So in the end the GDI was chosen to provide an overall measure of running quality with an easily understood scaling.
One of the critiques of previous articles is the focus on individual joints/planes without an overall measure of the activity. The additional analysis of 9 GVS variables is might add confusion rather that clarity. For example, it is quite possible that say the sagittal plane ankle GVS is significantly better for one task. For the overall GPS to be non-significant this means other GVS variables are worse. Yet several other GVS variables could be slightly worse but not statistically significant. In general, this study wanted to focus on a single overall measure of walking and running.
I have a number of other comments contained in the annotated pdf:
Revisions: From the PDF file, all comments have been revised. Summary of revisions provided below:
- Intro: these two sentences are somewhat confusing saying unilateral CP is like TD then that children with CP are not. Please reword. Revision: The two sentences referred to have been revised to clarify Page 2 top paragraph).
- Methods: I think your method section would be better ordered
Participants
Data collection
Gait deviation index (this follows logically as it is calculated post data capture)
Revision: Methods section re-ordered as suggested (pages 2 to 4)
- Methods: I think a better word here would be deviation as kinematics different to the mean do not necessarily infer pathology. Revision: Pathology revised to deviation (page 3, section 2.3, 1st paragraph).
- Methods: It was not apparent to me that the GDI* run was derived from calculation of the GPS and transformation to the GDI based on the strong relationship between the measures. This should be made clear in the text. Revision: Using much of the suggested text for the revision. The last paragraph of the GDI description has been revised (page 4, 1st paragraph).
- Methods: 8 reference in for GPS not GDI. Revision: Correct reference for GDI updated to Schwartz and Rozumalski now reference 11 with addition of several other citations.
- Methods: It would be interesting to know if these participants have had previous muscle or bony surgery. Response: Unfortunately, previous surgery was not collected as part of the ethics board approval, only that no orthopedic surgery occurred in previous 2 years.
- Participants: It is more appropriate to refer to the child with CP rather than patient with CP. Revision: the word patient revised to children. Note: the document was searched for patient and replaced with children or participant in multiple instances. Patient is not used in the manuscript any longer (multiple instances).
- Methods: use a time rather then 1 frame of data as this can vary do to capture frequency. Revision: revised to 0.01 seconds instead of 1 frame. (page 3, end of section 2.1)
- Methods: As two different systems were used please provide further detail on
- filtering of trajectories and force data
- what model or the details of model used in visual 3D software.
- Were any modifications made to Vicon PiG model
- Was data normalized to gait cycle?
- How many trials were collected?
Revision: Revisions to address these comments have been added to the Methods section, 2.2 Data collection as follows for the Erie center followed by the Spokane Center (page 3):
“Joint angles were obtained using a six degree of freedom model including 4 markers on pelvis, clusters of 4 markers on the thigh and shank segments, and foot and ankle markers [9]. Trajectories were filtered with a 4th order zero lag Butterworth, cutoff 10 Hz. Force plate data were not filtered; note kinetic data was not used in this study. Joint angle data were normalized to 0-100% of the gait cycle. Three successful trials (characterized by the foot falling completely within the bounds of the force platform without targeting) per limb were collected for walking and five per limb for running”
“The Plug-in-Gait model was modified by calculating the pelvic angles using a rotation, obliquity, tilt sequence (Baker et al.); in addition, patella (Wren et al) and tibial crest markers were employed instead of wands. Trajectories were also filtered with a 4th order zero lag Butterworth, cutoff 10 Hz. Force plate data were not filtered. Joint angle data were normalized to 0-100% of the gait cycle. Six successful trials (characterized by the foot falling completely within the bounds of the force platform without targeting) per limb were collected for walking and one per limb for running.”
- Methods: How was representative trial selected? Revision: Representative trial was selected randomly. Word ‘randomly’ added to sentence (1st sentence, section 2.4)
- Data analysis: level of what? Revision: “Statistical significance threshold was set at p < 0.05 for comparing means between Activity and Center.” (last sentence, section 2.4)
- Table 1: what do numbers out of brackets represent? Revision: Table re-arranged so that characteristics with counts appear at top, followed by parametric characteristics. Table 1 revised to state means with ± standard deviations and removing brackets as suggest by reviewer #3.
- Results: A quantitative measure of gait deviation for each of the kinematics variables rather than visual inspection of the gait kinematics would strengthen this article. Response: see overall response for rationale of focusing on overall gait deviation.
- Figure 1. Revision: caption revised as suggested.
- Discussion: Revision: “Foot strike patterns for barefoot running kinematics vary from shod running [13].” Was deleted as suggested. (page 8, top paragraph)
- Appendix B figures: Title graphs kinematics walking, kinematics running. Revision: Figure titles revised as suggested.

Reviewer 3 Report
The work is very well structured, and the topic is interesting.
1 . why didn't you use the same material on Erie and Spokane? Could the use of different materials have an influence on the results?
2. Table 1. Put in the legend the meaning of the measurements (m), (cm), (#).
I propose that you place yourself under Erei, Spokane and All (mean ± sd) and remove that information from the legend.
In this table it would also be interesting to place p of SHAPIRO (surely greater than 0.05) that you forwarded to ANOVA with parametric results.
3. Figure 1. I consider that the presentation in terms of the statistically significant results should be presented in another way. Pay attention to * and then say what it means in the caption.
4. Figure 2. There must be a table associated with this figure, which supports the results numerically, an average of the deviations of the% of the Gait Cycle can be made.
5. I think it would be very important to understand the therapeutic approach in each of the centers. Once there seems to be greater therapeutic efficiency in center A compared to B. As a reader the therapist seeks to know the key to success and this article does not give us that. The conclusions are that there are no significant differences in the quality of gait and running in patients with CP. But those at the Erie center have a higher walk and run GDI. WHY?
6. It would be interesting to use a control group.
7 Uma abordagem que fazem na discussão no paragrafo:
"The study included children who were GMFCS level I (ability to perform gross motor skills including running) and II (minimal ability to perform gross motor skills including running) [10,11]. In retrospectively searching for up to the last 30 patients with diplegic CP, 5 out of 25 participants at Erie, (3 GMFCS Level I and 2 Level II) and 3 out of 33 participants at Spokane (all GMFCS level II) did not meet the criteria for running and were excluded from the study."
This approach should be placed in one flow diagram of participants
8. Appendix. They must put the caption on pages 9 and 11
Author Response
Reviewer #3:
Comments and Suggestions for Authors
The work is very well structured, and the topic is interesting.
1 . why didn't you use the same material on Erie and Spokane? Could the use of different materials have an influence on the results?
Response: There is not a standard universally accepted model for analyzing gait or running. One of the reasons/strengths of analyzing the data from 2 centers (Erie and Spokane) is the use of different models. The result that despite different models, the findings are the same, strengthens the conclusion that walking and running are similar for children with CP.
- Table 1. Put in the legend the meaning of the measurements (m), (cm), (#).
I propose that you place yourself under Erie, Spokane and All (mean ± sd) and remove that information from the legend.
In this table it would also be interesting to place p of SHAPIRO (surely greater than 0.05) that you forwarded to ANOVA with parametric results.
Revisions: meaning of measurements spelled out in first column. mean ± sd notation added for appropriate cells. Legend deleted. Added statement in Table title that none of characteristics were significantly different between labs (p > 0.05)
- Figure 1. I consider that the presentation in terms of the statistically significant results should be presented in another way. Pay attention to * and then say what it means in the caption.
Revisions: Presentation of the statistically significant results are presented in a revised manner using a * and revision of caption.
- Figure 2. There must be a table associated with this figure, which supports the results numerically, an average of the deviations of the% of the Gait Cycle can be made.
Response: Figure 2 is intended to show that there is not a consistent pattern of running for children with Cerebral Palsy. It is not intended to be a quantitative analysis.
- I think it would be very important to understand the therapeutic approach in each of the centers. Once there seems to be greater therapeutic efficiency in center A compared to B. As a reader the therapist seeks to know the key to success and this article does not give us that. The conclusions are that there are no significant differences in the quality of gait and running in patients with CP. But those at the Erie center have a higher walk and run GDI. WHY?
Response: It is unknown why the GDI for walking and running is higher at the Erie center. It may not be the therapeutic approach. It might simply be the referral pattern of children with CP. It could be other factors. Therefore, the results focused on the lack of differences between walking and running at each center. The intent of the paper was to determine if there is a difference between how well a child with CP walks vs. runs. Understanding this information is the first step, and may then be used to improve the therapeutic / treatment interventions in this population.
- It would be interesting to use a control group.
Response: A control group of typically developing children different than the basis for calculating the GDI was not available. We expect that a control group of typically developing children would not be greater than 1 standard deviation from the mean between walking and running as they would be similar to the group that makes up for the basis of calculating GDI A control group of a diagnosis other than CP was also not available.
- "The study included children who were GMFCS level I (ability to perform gross motor skills including running) and II (minimal ability to perform gross motor skills including running) [10,11]. In retrospectively searching for up to the last 30 patients with diplegic CP, 5 out of 25 participants at Erie, (3 GMFCS Level I and 2 Level II) and 3 out of 33 participants at Spokane (all GMFCS level II) did not meet the criteria for running and were excluded from the study."
This approach should be placed in one flow diagram of participants.
Response: The suggestion for a flow diagram of participants is appreciated. Often the diagram indicates recruitment such as participants not interested in the study, etc. This was a retrospective study, a sample of convenience. Even with this sample of convenience, a few subjects did not achieve double float or running. These values were simply to add support to the results already found that running ability is not better than walking and in fact, a few subjects could not strictly achieve running.
- Appendix. They must put the caption on pages 9 and 11
Revision: Appendix has been added to the captions as suggested.

Reviewer 4 Report
Dear Authors,
I have some comments on your article:
- At the end of the Introductions section, there is no information on how the article is organized.
- The Conclusions section should be expanded.
- Literature should be checked if there are no newer items. Especially from the last 18 months. It would be good to add several references.
Best regards
Author Response
Reviewer #4:
Comments and Suggestions for Authors
Dear Authors,
I have some comments on your article:
At the end of the Introductions section, there is no information on how the article is organized.
Revision: Revisions have been added to the end of the introduction to indicate the organization of the article. Indicating kinematic data collection, GDI calculation, analysis, and results. (page 2, last paragraph of Introduction).
The Conclusions section should be expanded.
Revision: The conclusions have been revised and expanded. The additional findings related to age have also been included. (page 8, Section Conclusions).
Literature should be checked if there are no newer items. Especially from the last 18 months. It would be good to add several references.
Revision: A paragraph has been added to introduction including several studies related to running in children with cerebral palsy. Several references within the last 18 months have been added. This paragraph is now the 4th paragraph in the introduction, Page 2 (new paragraph inserted).
“Several characteristics have been reported to be altered in running for children with CP. During running, children with cerebral palsy have been shown to have reduced leg stiffness [7], altered dynamic stability (increased medio-lateral margin of stability) [8], and reduced ankle plantarflexor power compared to typical developing children in running [9]. Further, a running training program has been shown to increase ankle power generation and hip flexor power in swing [10]. These altered characteristics in running for children with CP are important, but do not compare the overall kinematics quality between walking and running.”
Best regards

Reviewer 5 Report
The aim of this study was to apply an overall score of gait consistency (Gait Deviation Index) to children with cerebral palsy to test the observation that they tend to run more like unaffected children even though their walking gait is demonstrably impaired. Overall, this is a well-written and well-designed study. The authors have been very thorough in their description of the study, in particular because they included data collected at two different sites. There were site-specific differences in the data which they considered in their analysis of the validity of their findings.
There are only two issues I feel need to be considered by the authors. First, in the second paragraph of the Introduction, they describe running for children with cerebral palsy as "a skill acquired later in development after the brain has healed". I am not sure that 'healed' is the best choice of words in this instance.
Second, the authors have chosen to include gait data curves (Figure 2) provided by Vicon. While this may be convenient for the authors, the figures tend to have fixed ranges on the y-axes, which is probably acceptable for gait data from an unaffected population. However, this will truncated data that is in any significant way different from so-called 'normal'. I suggest the authors create their own plots or, at the very least, edit the Vicon plots so that they autoscale the y-axes.
Author Response
Reviewer #5:
Comments and Suggestions for Authors
The aim of this study was to apply an overall score of gait consistency (Gait Deviation Index) to children with cerebral palsy to test the observation that they tend to run more like unaffected children even though their walking gait is demonstrably impaired. Overall, this is a well-written and well-designed study. The authors have been very thorough in their description of the study, in particular because they included data collected at two different sites. There were site-specific differences in the data which they considered in their analysis of the validity of their findings.
There are only two issues I feel need to be considered by the authors. First, in the second paragraph of the Introduction, they describe running for children with cerebral palsy as "a skill acquired later in development after the brain has healed". I am not sure that 'healed' is the best choice of words in this instance.
Revision: The word ‘healed’ deleted and the sentence simply states later in development (page 1, second paragraph).
Second, the authors have chosen to include gait data curves (Figure 2) provided by Vicon. While this may be convenient for the authors, the figures tend to have fixed ranges on the y-axes, which is probably acceptable for gait data from an unaffected population. However, this will truncated data that is in any significant way different from so-called 'normal'. I suggest the authors create their own plots or, at the very least, edit the Vicon plots so that they autoscale the y-axes.
Revision: Figure 2 has been revised to increase the range on the y-axis so data from participants is not truncated. The figures were originally created by the authors and the scales were selected in attempt to balance scaling. The figure has been revised thanks to your feedback.
Round 2
Reviewer 1 Report
The revision is an excellent improvement in the paper. I have happy for publication to proceed without further changes.
Author Response
No further revisions requested. Thank you for your time reviewing this paper.
Reviewer 2 Report
The authors present a revised manuscript comparing the deviation in walking verses running in children with cerebral palsy compared to typically developing children.
The authors have made significant revisions to address my concerns regarding the manuscript. I am happy to reccomend this article for publications. I have just a few minor comments.
The authors are obviously familalir with the GDI. However, I find it curious that they find a number that is standardised to 100, SD 10 easier to understand than the GPS which is simply a measure of how far a persons gait deviates from typically developing. For many without a technical background the mathematics of the GDI is beyond thier scope of knowledge.
Data collection, It is succifent to say force plate data was used for detection of foot strike and toe off events.
Table 1, it is still unclear that the values are mean +- std. this can be stated in caption.
I think the GDI vs Age graphs could go in the appendix.
Author Response
The authors present a revised manuscript comparing the deviation in walking verses running in children with cerebral palsy compared to typically developing children.
The authors have made significant revisions to address my concerns regarding the manuscript. I am happy to recommend this article for publications. I have just a few minor comments.
1. The authors are obviously familiar with the GDI. However, I find it curious that they find a number that is standardised to 100, SD 10 easier to understand than the GPS which is simply a measure of how far a persons gait deviates from typically developing. For many without a technical background the mathematics of the GDI is beyond their scope of knowledge.
Response: We thank the reviewer for their understanding of the GDI and GPS, clearly being familiar with the details of both indices. For the authors, we relate more easily to a given GDI, for example a score of 85, and rapidly determining it as 1.5 standard deviations from typical mean. The reviewer’s preference is also appreciated.
2. Data collection, It is sufficient to say force plate data was used for detection of foot strike and toe off events.
Revision: Data collection paragraph revised to state force plate data was used for detection of foot strike and toe-off events. Other force plate details were deleted. Lines 111-125.
3. Table 1, it is still unclear that the values are mean +- std. this can be stated in caption.
Revision: Caption revised to indicate values in table are mean +- standard deviation.
4. I think the GDI vs Age graphs could go in the appendix.
Revision: The GDI verse age graphs that were contained in Figure 3 have been moved to the Appendix. Revisions in text Lines 200-204 were also made to refer to Appendix.
Reviewer 3 Report
I noticed that the authors took into account all the proposed changes.
And the answers to the questions are satisfactory.
As such, I believe the article is fit for publication.
Thank you very much and, congratulations on all the work done.
Author Response

(The authors gave the same response as above.)

Reviewer 4 Report
Dear Authors,
Thank you very much for introducing changes that have improved the quality of the article.
I have no more comments.
Best regards
Author Response

(The authors gave the same response as above.)
